# A Manifest against the Homogenisation of Childbirth Experiences: Preserving Subjectiveness in a Large Dataset of the «Babies Born Better» Survey

**Mário J. D. S. Santos** [1,2,3,*] and **Dulce Morgado Neves** [3]

1   Comprehensive Health Research Center (CHRC), Universidade NOVA de Lisboa, 1600-560 Lisboa, Portugal
2   Public Health Research Centre, NOVA National School of Public Health, Universidade NOVA de Lisboa, 1600-560 Lisboa, Portugal
3   Centro de Investigação e Estudos de Sociologia, ISCTE-Instituto Universitário de Lisboa (ISCTE-IUL), 1649-026 Lisboa, Portugal; dulce_neves@iscte-iul.pt
*   Correspondence: mario.santos@ensp.unl.pt

**Abstract:** The Babies Born Better international project aimed at surveying women's experience in childbirth, privileging the qualitative description of this experience. It was translated into several languages and, in each country, there were different strategies for data analysis. However, analysing a qualitative dataset of this dimension, without completely transforming qualitative into quantitative data, poses practical challenges to researchers. Thus, in this article, we aim to explore the potential of using a qualitative data analysis software to avoid homogenising women's experiences and preserve the subjectivity of responses in the analysis of open-ended questions of the B3 survey. We focused on the Portuguese version of the survey, reporting a thematic, computer assisted qualitative data analysis of 1348 responses. The software acted as a mediator of the researchers' analysis and interpretation, beyond classical content analysis, without converting qualitative into quantitative data through plain word count. Exploring new possibilities of interpreting not only the meaning, but the relations between categories, may expand the scope of qualitative data analysis. However, we argue that the use of a software should not be overvalued, as such strategy should always remain as subsidiary to the researcher's subjective interpretation of data.

**Keywords:** Portugal; birth experience; maternity care; subjectivity; intersectionality; online survey; qualitative data analysis; CAQDAS



## 1. Introduction

In most European countries, and throughout the world, studies on birth conditions and maternal health reflect the dominance of a positivist paradigm, mainly based on quantitative methodologies and extensive analyses that tend to homogenise women as a group, with a clear emphasis on statistics and demographic data (Clausen and Santos 2017; Pintassilgo and Carvalho 2017). Despite some examples from the literature, little is systematically produced on women's subjective childbirth experiences as a way of assessing the quality of health care (Downe et al. 2018).

Critical perspectives on maternity care emerged in the 1970s, after decades of compliance between the social and the medical sciences. Following the critiques to the patriarchal matrix of health services and professional practice, several studies were dedicated to "giving the voice back to women", valuing the subjective dimensions of the experience of pregnancy and childbirth (Oakley 2016). Yet, assessing the quality of maternity care is still often restricted to the evaluation of objective indicators, such as the maternal and perinatal mortality rates (Miller et al. 2016; World Health Organization 2018). Not so frequently, obstetric intervention rates are also considered.

Childbirth is a complex phenomenon and adopting transdisciplinary and intersectional approaches may allow integrating both its objective and subjective dimensions. New perspectives reinforce the need to look beyond the limited scope of mortality rates, and focus on the subjective conditions of each childbirth, particularly when facing very low maternal and perinatal mortality rates (Pintassilgo and Carvalho 2017; Freedman and Kruk 2014). The current debate around obstetric violence as a form of violence against women has contributed to place the issue of disrespect and abuse in childbirth into the political agenda, framing it as a structural, worldwide problem (Freedman and Kruk 2014; Sadler et al. 2016). Likewise, in Portugal, a survey to 3833 women (APDMGP 2019) revealed an overuse of obstetrical interventions and several cases that could be said to configure obstetric violence. Following these results, the Portuguese parliament recently addressed 16 recommendations to the government regarding the improvement of the quality of maternity care and the promotion of women's rights in pregnancy and childbirth, including the development of a survey for assessing the satisfaction of women and health professionals regarding the provision of maternity care (Assembleia da República 2017).

Indeed, learning from the discourse of women on their childbirth experiences to inform the public and the political debates on the quality of maternity care is one of the possible approaches. As such, the Babies Born Better (B3) online survey was developed as a tool to collect data from a high number of women's individual and subjective childbirth experiences. B3 is a long-term project examining experiences of women who have given birth in the 5 years previous to the survey. It was first released online in 2013, in English and then translated and launched in more than 40 countries, receiving close to 40,000 responses until the end of 2015. The large qualitative dataset generated through this survey poses practical challenges to researchers. Natural language processing tools and machine learning technologies could aid the analysis of larger B3 datasets, but they require advanced computation skills that are readily available to few, if any, of the B3 research teams. Furthermore, these word-mining technologies require converting qualitative into quantitative data, which is precisely what we want to avoid. Yet, given the challenges of analysing such large datasets without these tools, it is not surprising that so little has been published so far reporting findings from the B3, despite being such a widespread international survey. To date, few journal articles were published reporting a completed analysis of national B3 data, namely articles from Croatia (Raboteg-Šarić et al. 2017), from Austria (Luegmair et al. 2018), and from Greece (Gouni 2020). The aim of this paper is thus to illustrate how the current tools of computer-assisted qualitative data analysis software (CAQDAS) can help the researchers grasping the analytical potential of a large and complex qualitative dataset like the one produced through B3 and, at the same time, preserving the subjectivity inherent to qualitative data analysis. Through the application of a qualitative approach to an extensive database, our intention is to counteract standardisation, giving way to more inclusive and intersectional perspectives, attentive to the diversity of perinatal experiences.

## 2. Materials and Methods

The B3 survey was disseminated through virtual communities and social media. The first Portuguese edition was launched in 2014. When the B3 website was opened, it provided a portal for accessing the questionnaire in the chosen language. Respondents could choose any language regardless of their country of residence.

Ethics approval for this international study was granted by the Ethics Committee of the University of Central Lancashire (UCLAN) in the UK (Ethics Committee BuSH 222). All data, in digital or hard copy form, was stored and handled in accordance with the General Data Protection Regulation, the UK Data Protection Act (1998), as well as the University of Central Lancashire (UK) guidelines. The front page of the survey offered information about confidentiality, consent, and the scope and aims of the survey. Being an online survey, all participants consented to participate by choosing to progress beyond this front page.

The survey had two sections; the first collecting demographic and health information, producing mostly quantitative data; the second with open-ended questions about the experience of childbirth, allowing the production of qualitative data. In this second section, instead of using predefined options, each woman could use her own words to describe the positive and negative dimensions of her experience of childbirth. We will focus on this second section of the B3 survey and describe the process of using CAQDAS for content analysis of the Portuguese B3 results, as a way of illustrating some of the potentialities within this strategy.

In the case of the B3 data, some aspects of the use of CAQDAS must be highlighted. Being an online survey, there is no direct interaction between researcher and informants, which eliminates the possibility of using the situations of data production and transcription as the first stage of analysis (Evers 2011). On the other hand, given the relatively high number of responses to the Portuguese version—1348 valid responses to the second section of the survey—opting for a classical content analysis would prove to be difficult and time-consuming, and it is arguable if it would, in fact, improve the quality of the analysis. For this analysis, MaxQDA® was the elected software.

Content analysis allows seeking the social meaning underneath the description of a certain phenomenon, intertwining context, data, and theory, to contribute to the knowledge of that same phenomenon (Guerra 2006; Krippendorff 2004). Flexibility is an advantage, as it adds value to the role of the researcher in data production and analysis. Each analysis is a unique research process. Acknowledging the strengths of the subjectivity of each interpretation can be more productive than seeking an over-standardisation of the analysis for the sake of transferability (Graneheim and Lundman 2004).

So far, among the completed and published qualitative data analysis of the B3 survey, the research teams from Austria and from Croatia report relatively small datasets. Raboteg-Šarić and colleagues (Raboteg-Šarić et al. 2017) report findings from a sample of 341 women who gave birth in Croatia, while Luegmair and colleagues (Luegmair et al. 2018) analysed 539 Austrian responses. Gouni had a larger sample, with 2089 Greek women, but the analysis was mostly quantitative (Gouni 2020). Yet, in Portugal, between April 2014 and September 2015, the B3 survey received a total of 1348 valid responses. This represented a much larger amount of quantitative and, more importantly in this case, qualitative data that posed some challenges to a timely completion of data analysis.

The second section of the B3, in which this article is focused, had four questions:

1. What were the three best things about the care you got there? Please put the very best thing at the top of the list. [Allowed 3 answers]
2. If you had the power to make three changes in the care you had, what would the changes be? Please put the most important change at the top of the list. [Allowed 3 answers]
3. Imagine a very close friend or family member is pregnant. They have asked you to give them a really honest description of the care you got at the place where you had your last baby. You can only use up to six words or phrases. What would those words or phrases be? [Allowed 6 answers]
4. Please write any comments you want to make here. These could explain your answers in more detail, or add any other information you would like us to know about your experiences with maternity care. [Allowed 1 answer]

These questions are further referred to as (1) positive aspects, (2) negative aspects, (3) experience description, and (4) comments.

The analytical strategy was reframed as more familiarity with the data was gained. Raboteg-Šarić and colleagues (Raboteg-Šarić et al. 2017), for example, report the use of a classical approach to coding and analysis of the Croatian responses to the B3. However, this was not possible in the case of the Portuguese dataset, given the dimension of the sample. Luegmair et al. (Luegmair et al. 2018), for the analysis of the Austrian B3 dataset, opted for a deductive coding framework based on the existing literature on childbirth experiences.

For the Portuguese case, when coding, we wanted to privilege an inductive approach, where codes and categories emerged from the experiences reported by women. A simple quantitative approach, through word counting, had major limitations: the context in which the words were used was not taken into account; words with related meaning were not aggregated, for example, obstetrician, doctor, Dr., physician, and gynaecologist. Further exploration of the adequacy of the CAQDAS tools led to the definition of the final analytical strategy (Figure 1).

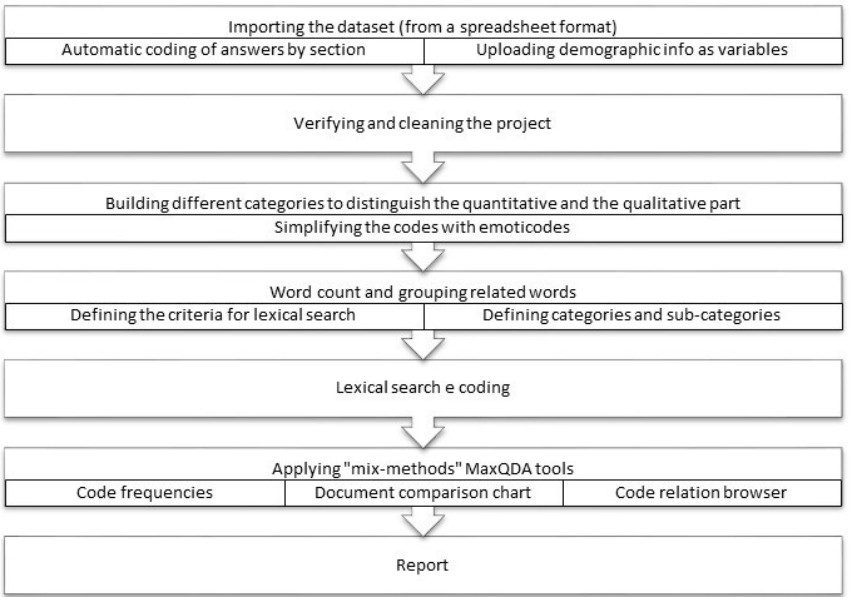

**Figure 1.** B3 qualitative data analysis strategy.

The original dataset was a structured spreadsheet, which allowed the automatic importing and coding of each paragraph by question. As such, each response in the survey generated one document within the MaxQDA project, and every document created had the same answer in the same paragraph. Demographic and health information from the first section of the survey were also imported as document variables (VERBI Software 2015). A selection of variables was also included as text in each document, working as a visual aid to help contextualise the interpretation (age, country of residence, language, closest city, number of children, year of the last birth, place of birth, and the group of professionals who provided most of the care).

The automatic coding of the project—through which each answer is automatically given a different code according to the corresponding survey question—was then executed, verified, and the dataset was cleaned. This allowed for each response to be clearly identified with each question. When performing this automatic coding, we chose to code empty cells; but this proved to be disadvantageous and was undone. Some respondents gave no answer to the qualitative section of the survey and were excluded, with 1348 of the initial 1678 remaining. Ten women answered the survey in another language (6 in English, 3 in Spanish, and 1 in Bulgarian), and their answers were translated with an online tool (Google Translate). To better distinguish the initial automatic codes from the following coding for content analysis, automatic coded segments by question were labelled with symbols (the MaxQDA "emoticodes") while all other thematic codes were labelled with a code name and colour.

After this first stage, the analysis was performed. Most answers had only one word or a simple phrase, particularly for the two first questions. As such, a simple word count identified the most frequent words and similar words were grouped (e.g., option, no option, choice, preference). Within this list, several keywords were tested for further lexical search adequacy. After selecting the words or expressions with higher frequency—

and the corresponding keywords—a lexical search was used to code segments. Some words were also included due to their analytical relevance, regardless of their frequency, such as the ones related to obstetric interventions, and to less frequent professionals, e.g., paediatricians. Codes were organised in three main categories: components of the care received; interventions and practices; professionals. Each had several subcategories. Paragraphs, rather than individual words, were coded, to allow the interpretation of the meaning of each word in accordance to the segment where it was used. Content was checked and inadequate coding of words or meanings was excluded.

Last, based on this coding, we used some of MaxQDA analytical tools that seemed fit to the characteristics of the B3 datasets: code frequency, document comparison chart, and code relation browser. The outputs of these analytical steps could then be used to aid the researchers' interpretation and to illustrate the results in any given report.

## 3. Results and Discussion

CAQDAS brought new possibilities for content analysis. It enabled an inductive, classical-like content analysis by assisting the systematic coding of data and simplifying the access to a specific coded segment. Additionally, some of the tools made available through CAQDAS would be otherwise difficult to operationalise (Evers 2011). Nonetheless, there have been some critiques to the use of CAQDAS, arguing that it is a fallacious attempt to perform content analysis under a quantitative paradigm, and that it introduces analytical inflexibility inadequate to qualitative approaches (Gobo 2005). It is our understanding that, with CAQDAS, as with any other analytical tool, it is part of the role of the researcher to ensure the quality of the analytical process.

In the particular case of the Portuguese B3 qualitative dataset, the sample had 1348 women who have had a baby 5 years prior to completing the survey. The age ranged from 18 to 48 years, with a mean age of 35 and a modal age of 33. The hospital was the main place of birth (94.9%), but a considerable number of women reported a home birth (4.5%). Although limited, the official national statistics report around 1% of home births (Santos and Augusto 2016). As such, there seems to be an overrepresentation of home births, which further ahead proved to be useful. As mentioned above, the B3 survey was disseminated online through social media and, therefore, we are dealing with a convenience sample (Sue and Ritter 2007). This may explain the overrepresentation of home births, since this phenomenon has been reported in other studies on childbirth experiences in Portugal based on big non-probabilistic samples (APDMGP 2019). In this sense, women with unconventional birth experiences, such as those who gave birth at home, may feel particularly motivated to share their "exceptional" experiences and to participate in the survey.

All Portuguese regions were covered. Most births (83%) were full-term and most women (67%) mentioned they had no problems during pregnancy. Roughly half of all women (47.7%) declared having received most of the care during labour and birth by a combination of doctors and midwives or nurses. Approximately one third (31.3%) identified midwives and nurse-midwives as having provided most of the care, which makes them the most singularly represented professional group, followed by doctors (16.4%), nurses (2.9%), and others (1.7%). Once again, this fact reinforces the skewness of our sample, differing from the trend shown by official (2011) data on birth in Portugal, being the presence of doctors at delivery (67.4%) higher than the presence of obstetrical nurses (32.1%) or non-obstetrical nurses (0.4%) (Pintassilgo and Carvalho 2017).

Within the qualitative component of the dataset, each woman could use up to three words or sentences for describing the best things and three more for describing the things they would have changed. Yet, the average number of answers for describing the best things was higher (2.7 answers, compared to 2.0 answers to the negative aspects). Pregnancy and childbirth are still socially considered to be joyful occasions, even when they are not experienced as positive. Previous studies (César et al. 2018) show how women tend to share the positive aspects of their motherhood experiences more easily, consigning the

negatives to private circles. With pregnancy and childbirth, the same seems to happen, thus contributing to the optimistic imaginary around these events.

Furthermore, the idea that surviving is not enough to obtain positive maternal health outcomes is rather recent, and the importance of promoting positive birth experiences seems to have only recently caught the attention of international health authorities and organisations (White Ribbon Alliance et al. 2015; World Health Organization 2014, 2018). The prevalence of these social factors might be one of the reasons for these differences in the average number of responses per question.

To illustrate the potential of the use of a CAQDAS to preserve analytical subjectivity in B3 datasets or similar projects, we will focus on three additional strategies that aim to demonstrate the advantages of going beyond the plain word frequency analysis of this dataset: associating subcategories; interpreting beyond frequency; comparing response patterns.

### 3.1. Associating Subcategories

The description of the care received during labour and birth helps capture a general picture of women's experience in the sample. Attentiveness, support, friendliness, professionalism, and respect are the more frequently mentioned components of care. However, by combining each two components, we can see some of the combination of words that are frequently mentioned together in the same answer, e.g., choice and respect, attentiveness and friendliness, respect and the acceptance of what is natural, respect and information, and information and choice (Figure 2).

| | Support | Attentiveness | Friendliness | Respect | Kindness | Comfort | Humanisation | Security | Availability | Information | Natural (acceptance of) | Being alone | Privacy | Choice | Professionalism | Trust | Freedom |
|---|---|---|---|---|---|---|---|---|---|---|---|---|---|---|---|---|---|
| Support | | 6 | 3 | 8 | 5 | 4 | 7 | 4 | 3 | 9 | 6 | 3 | 1 | 9 | 2 | 4 | 2 |
| Attentiveness | 6 | | 19 | 6 | 11 | 1 | 7 | 3 | 10 | 3 | 2 | 2 | 0 | 3 | 6 | 1 | 1 |
| Friendliness | 3 | 19 | | 0 | 4 | 0 | 3 | 0 | 7 | 1 | 2 | 1 | 0 | 2 | 3 | 0 | 0 |
| Respect | 8 | 6 | 0 | | 1 | 2 | 12 | 8 | 1 | 16 | 18 | 2 | 6 | 31 | 3 | 2 | 7 |
| Kindness | 5 | 11 | 4 | 1 | | 1 | 3 | 0 | 2 | 1 | 0 | 1 | 1 | 0 | 0 | 0 | 0 |
| Comfort | 4 | 1 | 0 | 2 | 1 | | 1 | 4 | 0 | 2 | 1 | 1 | 4 | 5 | 0 | 1 | 0 |
| Humanisation | 7 | 7 | 3 | 12 | 3 | 1 | | 2 | 3 | 8 | 6 | 1 | 0 | 6 | 0 | 0 | 1 |
| Security | 4 | 3 | 0 | 8 | 0 | 4 | 2 | | 0 | 5 | 0 | 1 | 2 | 7 | 4 | 10 | 2 |
| Availability | 3 | 10 | 7 | 1 | 2 | 0 | 3 | 0 | | 4 | 2 | 1 | 1 | 2 | 0 | 1 | 0 |
| Information | 9 | 3 | 1 | 16 | 1 | 2 | 8 | 5 | 4 | | 5 | 3 | 0 | 15 | 0 | 1 | 4 |
| Natural (acceptance of) | 6 | 2 | 2 | 18 | 0 | 1 | 6 | 0 | 2 | 5 | | 1 | 0 | 13 | 0 | 1 | 1 |
| Being alone | 3 | 2 | 1 | 2 | 1 | 1 | 1 | 1 | 1 | 3 | 1 | | 2 | 1 | 1 | 0 | 0 |
| Privacy | 1 | 0 | 0 | 6 | 1 | 4 | 0 | 2 | 1 | 0 | 0 | 2 | | 1 | 0 | 0 | 1 |
| Choice | 9 | 3 | 2 | 31 | 0 | 5 | 6 | 7 | 2 | 15 | 13 | 1 | 1 | | 2 | 5 | 14 |
| Professionalism | 2 | 6 | 3 | 3 | 0 | 0 | 0 | 4 | 0 | 0 | 0 | 1 | 0 | 2 | | 4 | 0 |
| Trust | 4 | 1 | 0 | 2 | 0 | 1 | 0 | 10 | 1 | 1 | 1 | 1 | 0 | 5 | 4 | | 0 |
| Freedom | 2 | 1 | 0 | 7 | 0 | 0 | 1 | 2 | 0 | 4 | 1 | 0 | 1 | 14 | 0 | 0 | |

**Figure 2.** Association of two components of care in the same answer.

As mentioned earlier, many answers had only one word. The low number of answers where a combination of these words is mentioned is, thus, not unexpected. Looking, for example, at the responses to questions 3 (experience description) and 4 (comments), we can see some of these associations:

Respect and choice:

- "The doctors I met were highly disrespectful, they always tried to raise fear in my options, and even refused to give me the care I asked for . . . " (Porto, 33 years, home birth)
- "My son's birth was terrible . . . labour was progressing normally, they were respecting everything, until the obstetrician entered the room, he was in haste to go home, and he humiliated me and forced me to break my waters against my choice." (Viana do Castelo, 35 years, hospital birth)

### 3.1.1. Attentiveness and Friendliness

- "It was a really easy birth, non-instrumented, performed by a female intern, very friendly and attentive. In general, it was a marvellous experience." (Funchal, 34 years, hospital birth)
- "Nurses were extremely friendly and attentive." (Lisbon, 28 years, hospital birth)

### 3.1.2. Respect and the Acceptance of What Is Natural

- "This is a hospital that respects natural birth, and women." (Almada, 35 years, hospital birth)
- "It was not as natural as I would like, but at least they respected my choice of not having a caesarean section. (Lisbon, 36 years, hospital birth)

### 3.1.3. Respect and Information

- "[I could have had] more humanised care, by informing about what they will do, and what is happening, and respecting the parents' will." (Lisbon, 34 years, hospital birth)
- "I didn't like the fact that one of the doctors attending me, not only was he really blunt and disrespectful, to me and to the father, but he also pressed my belly without even informing me." (Portimão, 35 years, hospital birth)

### 3.1.4. Information and Choice

- "They informed me before any procedure, they gave me freedom of choice in many aspects, I could choose the position to give birth." (Matosinhos, 31 years, hospital birth)
- "In the place where I had my baby, I felt they didn't give me that much choice and information. It's all pretty much protocol-based and they don't consider the individual person. I felt I could have been better informed, that they could have listened to me more and respected my rhythm." (Funchal, 26 years, hospital birth)

Understanding how respondents combine words in the description of their experiences allows us to go further in capturing the meanings attributed to such contexts. In this sense, although the metric is not the most important aspect in a qualitative analysis, it is still elucidative to realise that the association between respect and choice is the one that obtains the highest frequency in the description of the experiences of childbirth, in line with results from other studies (Downe et al. n.d., 2018).

### 3.2. Interpreting beyond Frequency

The evolution of birth conditions in Portugal reveals a trend of institutionalisation, with a strong presence of doctors attending childbirth (Pintassilgo and Carvalho 2017). Still, as mentioned earlier, nurses and midwives form the singular professional group that was more frequently identified by respondents as having provided most of the care. However, nurse-midwives are commonly designated by "nurses". Thus, it is not possible to be sure if women were referring to nurses, to nurse-midwives, or to midwives, when using the word "nurse". As such, these related words were coded as "nurses or midwives". When looking at the positive and negative aspects, this is also the most relevant professional group, closely followed by doctors (Figure 3).

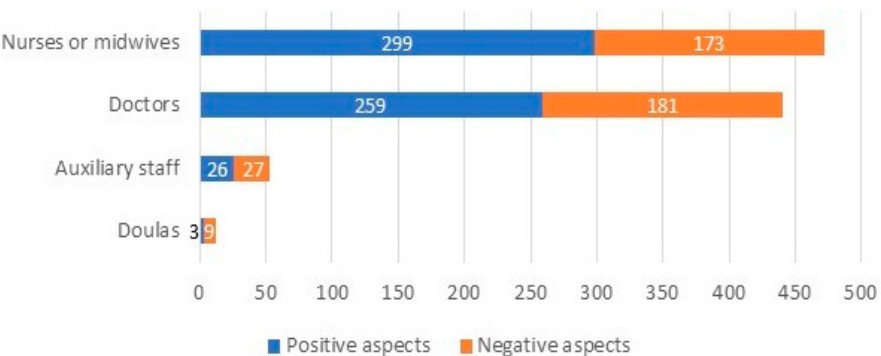

**Figure 3.** Professionals as part of the positive or negative aspects of childbirth.

Both doctors and nurses or midwives were mainly mentioned as part of the positive aspects of childbirth. Regarding the most mentioned group, in a total of 472 mentions, nurses or midwives were referred 299 times associated with positive aspects of the experience, and 173 times associated with negative aspects. Auxiliary staff and doulas were also mentioned, but less frequently. While the auxiliary staff has around half of its references in each side of the experience, doulas are mainly mentioned associated with the negative aspects.

With Figure 3, we aim to demonstrate why a simple conversion from qualitative to quantitative data may lead to imprecise conclusions. To correctly interpret these findings, we must move beyond the plain frequency of subcategories. With a CAQDAS, however, we can easily explore the content of each code and category, making it simpler for the researcher to return to the discourse of the respondents and to consider the meanings of these categories and subcategories in the analysis. Coding the reference to a professional as part of the positive or the negative aspects of the childbirth experience did not always represent that a professional conduct or intervention were perceived as positive or negative by each respondent. Particularly regarding this distribution of professionals, it is worth mentioning that in few yet relevant cases, institutional limitations were mentioned as negative (e.g., the lack of midwives and nurses in the hospital was perceived as negative) and not the practice of this professional group; some women referred to personal options they later regretted (e.g., having a friend paediatrician at a home birth, which ended up interfering with the desired calm atmosphere). Doulas stand as a particularly interesting example. By looking back at the responses, we see that only one of the coded segments is directly related to the doula's performance, stating "[I wish] the doula would have stayed a couple of more hours after birth." (Lisbon, 36 years, home birth). All other negative references to doulas relate to the limits imposed by the hospital setting regarding the presence of doulas, with some women referring they had to choose between having their partner or their doula, but not both. These examples demonstrate how the importance of content analysis prevails in reading and interpreting these quantitative results. When converting qualitative into quantitative data, CAQDAS enables an easy return to the qualitative data and to the meaning behind each category, allowing us to look at these figures with caution, keeping in mind the importance of subjectivity and reflexivity for the interpretation of what these figures mean.

### 3.3. Comparing Response Patterns

Another strategy used to analyse the B3 Portuguese dataset is based on the comparison of response patterns between subsamples, in this case regarding the place of birth. Acknowledging the above-mentioned limitations of an analysis based on word count or subcategory frequency, this strategy allows a mid-range analysis and interpretation that does not compromise the subjectivity of each response.

Place is one of the many factors influencing the lived experience of giving birth (Birthplace in England Collaborative Group 2011; Burns 2015; Coxon et al. 2017). To analyse how each category (components of care, interventions, and professionals) varied in the

women's description of her experience, in three different places of birth—home, private hospital, and public hospital—three subsamples were created. From those who provided information on the place of birth, we selected a subsample with the most frequent private hospital (*n* = 67), and another with the most frequent public hospital (*n* = 98). All cases of home births formed the third subsample (*n* = 60). Given the fact that, as mentioned previously, all documents are structured, the document comparison chart was the selected tool for this analysis, and Figure 4 presents a fragment of the output files from this MaxQDA tool. Each frame represents a different place of birth. Each line corresponds to a respondent. Each cell represents an answer, and all coded segments have a colour per category. The first column represents the positive aspects, the second column represents the negative aspects, and the third and fourth columns represent the general experience description, and comments. From the outputs of this document comparison chart, we selected a fragment containing 50 responses that can be said to illustrate and to meaningfully represent the distribution of categories in each subsample. Grey cells are un-coded answers that do not fit within any of the mentioned categories. Empty cells represent blank answers.

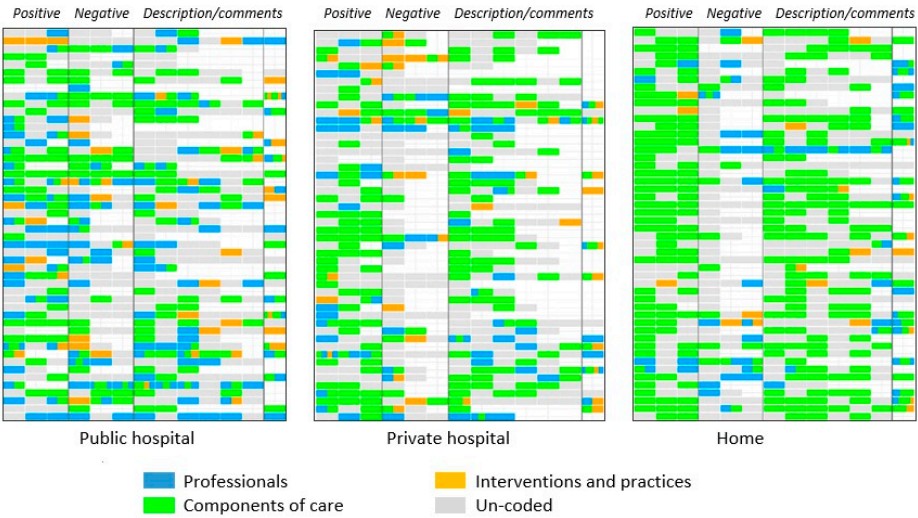

**Figure 4.** Comparing the distribution of categories by place of birth.

This comparison is useful as it immediately allows for a visual assessment of different response patterns between birth places. The experience of childbirth in this public hospital was reported referring to all categories, with no evident pattern. Professionals and interventions were mentioned both as part of the positive and the negative dimensions of care. The experiences of childbirth, in this subsample, seem to be reported taking all these factors into account.

In this private hospital, the distribution of categories is similar to the one in the public hospital; yet, overall, there seems to be less answers on the negative aspects of the experience of childbirth. There are more references to interventions—particularly as part of the negative aspects—and less references to professionals. Un-coded segments seem more common, as the selected categories do not encompass its content.

The picture of home births is quite distinct. The experiences were reported with few references to professionals and interventions. The components of care are clearly the dominant category, especially as part of the positive aspects. Most women gave only one answer regarding the negative aspects, but they remain mainly un-coded. It seems the experiences of home birth captured through this survey are largely reported referring to the circumstances and the atmosphere at home, and not that much referencing professionals or interventions. These differences in the response patterns between birth settings constitute an innovative set of findings that, to our knowledge, has not yet been discussed in the literature. The use of MaxQDA seems particularly fit for this sort of analysis of the B3 data and similar surveys.

## 4. Conclusions

The experience of childbirth is subjective and reporting such experience can be highly diverse. In this case, we wanted to perform an analysis of a big and complex qualitative dataset, establishing a compromise between extension and depth, and, thus, enhancing the perception of processes of differentiation. The analytical strategy presented in this paper, favouring the preservation of subjectivity and avoiding the homogenisation of birth experiences, is—we argue—a necessary stage towards an intersectional approach. The option of using CAQDAS for the analysis of the qualitative component of the B3 survey results seemed to be an adequate strategy. Yet, using CAQDAS to support our analysis, we arrived at a set of findings similar to what other researchers found when analysing smaller datasets without the support of CAQDAS (Downe et al. n.d.). Beyond classic content analysis and word counting, the software truly acted as a facilitator of the researchers' analysis and interpretation of subjective data.

Our analytical strategy revealed that the relationship between women and childbirth professionals is a key factor, contributing to the overall childbirth experience. The role of professionals emerged as being of great importance, configuring other aspects of women's experiences. All other captured dimensions somehow orbit these relationships. Going beyond a quantitative approach to this dataset also revealed how the place of birth seems to influence not only each childbirth experience, but also the way women describe their experiences, and the content of that description. Although there are some differences between the experiences described in public and private hospitals, the contrast is more evident when comparing hospital births with home births. In hospital births, specific professionals and interventions are more often mentioned, while in home births women seemed to prefer to describe components of care.

However, it is important not to over-quantify qualitative data. The quantitative processes supporting part of the analysis with a CAQDAS should not be overvalued and simply converting qualitative into quantitative data should be avoided. It is our understanding that the analysis with CAQDAS should remain subsidiary to the interpretation of the meaning within each answer. Exploring new possibilities of interpreting not only the meaning, but the relations between categories expands the scope of qualitative data analysis, allowing to go further in capturing diversity. The use of CAQDAS, as described, proves to be highly recommended to the analysis of B3 or any similar dataset.

Often, in extensive surveys, little attention is paid to qualitative information, usually collected through open-ended questions. Thus, the relevance of this work also lies in proposing a solution to counteract the inattention to qualitative data in big datasets, which poses ethical and methodological challenges to researchers. Particularly to the B3 project and to B3 research teams in other countries, this paper may provide an alternative to classical or quantitative content analysis.

In this analysis, only groups of words with higher frequency were included, and we acknowledge the limitation of having many relevant answers un-coded. A more comprehensive analysis and coding could have provided different outcomes and further insight about its content. Ultimately, the use of a CAQDAS should catalyse and not hinder the analytical process. Assessing the adequacy of such a tool is, and should continue to be, a subjective process in the hands of the researcher.

**Author Contributions:** Conceptualization, M.J.D.S.S.; methodology, M.J.D.S.S.; validation, M.J.D.S.S. and D.M.N.; formal analysis, M.J.D.S.S. and D.M.N.; investigation, M.J.D.S.S.; data curation, M.J.D.S.S. and D.M.N.; writing—original draft preparation, M.J.D.S.S.; writing—review and editing, M.J.D.S.S. and D.M.N.; visualization, M.J.D.S.S.; supervision, M.J.D.S.S.; project administration, M.J.D.S.S.; funding acquisition, M.J.D.S.S. and D.M.N. All authors have read and agreed to the published version of the manuscript.

**Funding:** This research was funded by COST—the European Cooperation in Science and Technology, grant number IS1405; and by Fundação para a Ciência e a Tecnologia, grant number SFRH/BD/99993/2014 and SFRH/BPD/94537/2013.

**Institutional Review Board Statement:** The study was conducted according to the guidelines of the Declaration of Helsinki, and approved by the Ethics Committee of the University of Central Lancashire (UCLAN) (Ethics Committee BuSH 222).

**Informed Consent Statement:** Informed consent was obtained from all subjects involved in the study.

**Data Availability Statement:** The data presented in this study are available on request from the corresponding author. The data are not publicly available due to ethical reasons.

**Conflicts of Interest:** The authors declare no conflict of interest.

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
