# Peer review of "A Manifest against the Homogenisation of Childbirth Experiences: Preserving Subjectiveness in a Large Dataset of the «Babies Born Better» Survey"

_socsci, doi:10.3390/socsci10100388_

Round 1

Reviewer 1 Report

The paper presents itself as a methodological work that is a “manifest against homogenisation”.It intends to show that CAQDAS can be used with large samples without falling into a purely quantitative analysis. For this, authors resort to open answers to an online questionnaire survey, and present three types of quantitative and quantitative analysis.The paper has potential, but it needs some corrections and developments in order to become publishable.

Specific comments follow.

The title is quite appealing, but it seems to promise more than it can deliver. Especially because one cant understand who are the "homogenizers".

In the introduction, lines 23 to 26, the authors argue that there is a “dominance of a positivist paradigm”, however, the works cited to support this argument were an ethnographic paper and a paper based on official statistics, an argument of this order is expected to be supported with a literature review or a more extensive state of the art.

Point 2. Materials and Methods. It is necessary to explain how the B3 team disseminated the questionnaire.

Line 89, authors can tell right away how many answers got in the questionnaire, instead of saying it was a “high number”, the suspense is unnecessary.

Line 97 – 98, authors argue that they privilege the subjectivity of interpretations to the detriment of seeking transferability, as if both were opposite situations, the results of qualitative studies can be “transferred” to other studies, since it does not imply that subjectivity always brings something new.

Line 104-105, the number of 1678 is not the final sample. At this point, authors can immediately present the size of the final sample and consider this, otherwise the value is inflated with cases that were not treated.

Line 129, authors argue that they followed an inductive method. However, when reading the paper, it is easy to understand that some decisions were taken guided by some theories/a prioris, for example, the decision to take the place of birth as a structural variable.

In the line 178 (among others) it is stated that CAQDAS is a mediator, but without specifying between intervening parties, the logic of a mediator is that it is “in the middle” of something, between two parties.

Line 191-192, there is an overrepresentation of home births compared to the official statistics figures. Could this be due to the survey's dissemination strategy? Or was it sought in order to have a more represented sample in this segment? The way it is stated, it seems that it was pure chance, but it is unlikely that it was.

Line 207-209 refer "Pregnancy and childbirth are still socially considered to be joyful occasions, even when they are not experienced as positive." This statement is extremely important to analyse the results that were classified, exactly, as “positive” and “negative”. However, this powerfull statement quickly jump to good practices proposed by international organizations. These are quite different issues and the first argument can very well be developed based on some literature that even exists specifically for the Portuguese case, such as for exemple: César et al 2018 “”To Suffer in Paradise”: Feelings Mothers Share on Portuguese Facebook Sites ”.

Line 217-220 in particular, but throughout the text. The authors make references to intersectionality without any definition of the concept or what they understand by intersectionality. There is just one reference, when the authors state that use intersectionality as a method and as a heuristic tool, which, in their words, combines individual and structural experience. This is a very abusive interpretation of the concept, which, although it counts as a plurality of interpretations, hardly escapes the idea that exists situations of inequality/oppression that are aggravated in conditions of an overlap of several social categories associated with lower status/ power. If the authors want to continue to have the term/concept of intersectionality in the title and throughout the paper, there needs to be some theoretical connection to the concept and some application to the data. Can use, as indicators of "intersectable categories", social status indicators that probably exists in the B3 questionnaire, such as occupation, education level, individual position in the social class structure, place of residence, nationality or place of birth. This is probably the biggest limitation of the paper, but sure it is surmountable.

Conclusion. What is the future direction of the research? Will this proposed qualitative data analysis allow for a more theoretical paper? What other authors with large datasets with qualitative data (or questionnaires with open questions) can learn and reuse from this study.

Reviewer 2 Report

This paper explores the use of CAQDAS software to assist in analysing large-scale qualitative data. The stated objectives in this study of promoting positive birth experiences and exploring this issue from an intersectional perspective are highly warranted. However, I think this study falls short in achieving these goals.

The authors suggest three ‘strategies’ for overcoming the limitations of simple word counting when converting qualitative to quantitative data – associating subcategories, interpreting beyond frequency and comparing response patterns. However, none of these strategies provide any substantial insight about the subjective birth experiences of women in Portugal. The most insightful information provided in this paper are the examples of quotes from respondents, which is a classic qualitative approach. Therefore, it is unclear how the CAQDAS software assists in preserving subjectiveness in this case. There may be alternative software or analytic approaches to the problem of analysing large-scale qualitative data. For example, the User-Generated Content (UGC) method proposed by Qian et al. (2020)*, may be a better fit for this purpose.

Second, the authors mention that “This approach brings us close to a conception of intersectionality as a method and an heuristic device” (lines 217-218). However, it is unclear how the proposed strategies achieve that, given that the use of CAQDAS is so limited in providing any information beyond general themes, whether these themes are mentioned in a positive or a negative context and how they correlate with one another. Furthermore, there is no analysis of the ethnic or socioeconomic background of the respondents in relation to their birth experiences.  

Overall, I agree with the authors objectives, though unfortunately, I do not see how the present study contributes toward these goals.  

*Qian, Y.; Liu, X.-y.; Fang, B.; Zhang, F.; Gao, R. Investigating Fertility Intentions for a Second Child in Contemporary China Based on User-Generated Content. Int. J. Environ. Res. Public Health 202017, 3905. https://doi.org/10.3390/ijerph17113905      

Author Response

Comment

Response

Changes

This paper explores the use of CAQDAS software to assist in analysing large-scale qualitative data. The stated objectives in this study of promoting positive birth experiences and exploring this issue from an intersectional perspective are highly warranted. However, I think this study falls short in achieving these goals.

Thank you for your review. As stated in the manuscript, “the aim of this paper is thus to illustrate how the current tools of computer-assisted qualitative data analysis software (CAQDAS) can help the researchers grasping the analytical potential of a large and complex qualitative dataset like the one produced through B3 and, at the same time, preserving the subjectivity inherent to qualitative data analysis.” In this sense, we believe we achieve these goals. We clarified the text throughout the manuscript so it becomes even more evident that this paper intends to demonstrate the usefulness of CAQDAS for analysing B3 data and other similar datasets. Promoting positive birth experiences and exploring this issue from an intersectional perspective are beyond the scope of this paper.

Changes were made throughout the text to assert the aim of the paper.

The authors suggest three ‘strategies’ for overcoming the limitations of simple word counting when converting qualitative to quantitative data – associating subcategories, interpreting beyond frequency and comparing response patterns. However, none of these strategies provide any substantial insight about the subjective birth experiences of women in Portugal. The most insightful information provided in this paper are the examples of quotes from respondents, which is a classic qualitative approach. Therefore, it is unclear how the CAQDAS software assists in preserving subjectiveness in this case. There may be alternative software or analytic approaches to the problem of analysing large-scale qualitative data. For example, the User-Generated Content (UGC) method proposed by Qian et al. (2020)*, may be a better fit for this purpose.

These strategies are illustrative of the potential of using CAQDAS in B3 or similar datasets, in line with the stated aim of this paper. Specifically, these strategies are presented as examples of how we could avoid a complete transformation of qualitative into quantitative data. Not only the subjectivity of responses but also the value of interpretation and the role of subjectivity in the analysis were intentionally preserved. Although analysing UGC is a relevant and valid strategy, the challenge here was to provide strategies for research teams that have lower levels of resources and are not familiar with machine learning or NLP tools. Changes were made in the manuscript to clarify these issues.

In the introduction:

The large qualitative dataset generated through this survey poses practical challenges to researchers. Natural language processing tools and machine learning technologies could aid the analysis of larger B3 datasets, but they require advanced computation skills that are readily available to few if any of the B3 research teams. Furthermore, these word-mining technologies require converting qualitative into quantitative data, which is precisely what we want to avoid. Yet, given these challenges of analysing such large datasets without these tools, it is not surprising that so little has been published so far reporting findings from the B3, despite being such a widespread international survey.

In the results and discussion:

To illustrate the potential of the use of a CAQDAS to preserve analytical subjectivity in B3 datasets or similar projects, we will focus on three additional strategies that aim to demonstrate the advantages of going beyond the plain word frequency analysis of this dataset: associating subcategories; interpreting beyond frequency; and comparing response patterns.

Second, the authors mention that “This approach brings us close to a conception of intersectionality as a method and an heuristic device” (lines 217-218). However, it is unclear how the proposed strategies achieve that, given that the use of CAQDAS is so limited in providing any information beyond general themes, whether these themes are mentioned in a positive or a negative context and how they correlate with one another. Furthermore, there is no analysis of the ethnic or socioeconomic background of the respondents in relation to their birth experiences. 

We agree with the comment and changes were made in the manuscript to better define our focus. The analytical strategy presented in this paper, favouring the preservation of subjectivity and avoiding homogenizing birth experiences, is - we believe - a necessary stage towards an intersectional approach. Nonetheless, we chose not to insist on the concept of intersectionality, considering that the respondents’ categories of differentiation were left out of the analysis.

Paragraph removed:

This approach brings us close to a conception of intersectionality as a method and an heuristic device [23], in the sense that it allows us to produce a multi-level analysis, taking into considerations both individual and structural dimensions of the experiences.

Added in the conclusion:

In this case, we wanted to perform an analysis of a big and complex qualitative dataset, establishing a compromise between extension and depth, and, thus, enhancing the perception of processes of differentiation. The analytical strategy presented in this paper, favouring the preservation of subjectivity and avoiding the homogenization of birth experiences, is - we argue - a necessary stage towards an intersectional approach.

Round 2

Reviewer 2 Report

The authors of the manuscript have addressed some of the concerns mentioned in the first round of review, including a refinement of the intersectionality argument. However, I am still in the opinion that the contribution of CAQDAS to maintaining subjectivity in qualitative analysis, at least as it is presented here, is quite limited. Thus, in order for the manuscript to be publishable, I would suggest a reframing of the advantages and disadvantages of CAQDAS in the context of the B3 survey:

For example, the two cases of “associating subcategories” and “comparing response patterns” do demonstrate ways in which the use of CAQDAS/ MaxQDA supports the process of qualitative data analysis. However, the example given for “interpreting beyond frequency” reveals a major limitation of the software, as it does not differentiate between negative aspects of professionals’ performance and the structural limitations on the provision of these services.

In sum, the manuscript requires a more accurate account of the strengths and weaknesses of CAQDAS in facilitating large-scale qualitative data analysis. Finally, some of the items in the list of references are missing.

Author Response

Table of revisions - 2nd round of revisions

Social Sciences

Manuscript socsci-1276872

Reviewer 2

Comment

Response

Changes

The authors of the manuscript have addressed some of the concerns mentioned in the first round of review, including a refinement of the intersectionality argument.

Thank you. The comments from the reviewers were quite helpful.

n.a.

The example given for “interpreting beyond frequency” reveals a major limitation of the software, as it does not differentiate between negative aspects of professionals’ performance and the structural limitations on the provision of these services.

In sum, the manuscript requires a more accurate account of the strengths and weaknesses of CAQDAS in facilitating large-scale qualitative data analysis.

This feature of the software, we argue, is a strength and not a limitation. In 3.2., we aim to demonstrate why a plan conversion from qualitative to quantitative data may lead to imprecise conclusions. With a CAQDAS, however, we can easily explore the content of each code and category, making it simpler for the researcher to return to the discourse and to the meanings of the respondents. We changed the text, for this argument to be clearer to the reader.

With figure 3, we aim to demonstrate why a simple conversion from qualitative to quantitative data may lead to imprecise conclusions. To correctly interpret these findings, we must move beyond the plain frequency of subcategories. With a CAQDAS, however, we can easily explore the content of each code and category, making it simpler for the researcher to return to the discourse of the respondents and to consider the meanings of these categories and subcategories in the analysis. Coding the reference to a professional as part of the positive or the negative aspects of the childbirth experience did not always represent that a professional conduct or intervention were perceived as positive or negative by each respondent. Particularly regarding this distribution of professionals, it is worth mentioning that in few yet relevant cases, institutional limitations were mentioned as negative (e.g. the lack of midwives and nurses in the hospital was perceived as negative) and not the practice of this professional group; and some women referred to personal options they later regretted (e.g. having a friend paediatrician at a home birth, which ended up interfering with the desired calm atmosphere). Doulas stand as a particularly interesting example. By looking back at the responses, we see that only one of the coded segments is directly related to the doula’s performance, stating “[I wish] the doula would have stayed a couple of more hours after birth.” (Lisbon, 36 years, home birth). All other negative references to doulas relate to the limits imposed by the hospital setting regarding the presence of doulas, with some women referring they had to choose between having their partner or their doula, but not both. These examples demonstrate how the importance of content analysis prevails in reading and interpreting these quantitative results. When converting qualitative into quantitative data, CAQDAS enables an easy return to the qualitative data and to the meaning behind each category, allowing us to look at these figures with caution, keeping in mind the importance of subjectivity and reflexivity for the interpretation of what these figures mean.

Finally, some of the items in the list of references are missing.

Thank you for noticing. The missing references were reintroduced. They were unintentionally removed.

References 1, 8, and 20 were reintroduced.